# Effects of Dietary Supplementation with Mulberry Leaf Powder on the Growth Performance, Lipid Metabolism Parameters, Immunity Indicators, and Gut Microbiota of Dogs

**DOI:** 10.3390/metabo13080918

**Published:** 2023-08-04

**Authors:** Aiying Yu, Cuiming Tang, Sutian Wang, Yuan Wang, Lian Chen, Zhiyi Li, Guoqing Luo, Jianwu Zhong, Zhengfeng Fang, Zhenjiang Wang, Sen Lin

**Affiliations:** 1Key Laboratory of Urban Agriculture in South China, Sericultural & Agri-Food Research Institute, Guangdong Academy of Agricultural Sciences, Guangzhou 510640, China; 2022214001@stu.sicau.edu.cn (A.Y.); tangcuiming@gdaas.cn (C.T.); wangyuan18789@163.com (Y.W.); luoguoqing@gdaas.cn (G.L.); zhongjianwu@gdaas.cn (J.Z.); 2Key Laboratory for Animal Disease Resistance Nutrition of the Ministry of Education, Animal Nutrition Institute, Sichuan Agricultural University, Chengdu 611130, China; zfang@sicau.edu.cn; 3State Key Laboratory of Livestock and Poultry Breeding, Guangdong Key Laboratory of Animal Breeding and Nutrition, Institute of Animal Science, Guangdong Academy of Agricultural Sciences, Guangzhou 510640, China; wangsutian@gdaas.cn

**Keywords:** dogs, mulberry leaf powder, lipid metabolism, gut microbiota, immunity

## Abstract

Overfeeding and a lack of exercise are increasingly causing obesity in dogs, which has become a big problem threatening the health of dogs. Therefore, it is necessary to investigate how dietary regulations can help to improve dogs’ body conditions and minimize obesity. This study was carried out to investigate the effects of dietary mulberry leaf powder (MLP) supplementation on the growth performance, lipid metabolism parameters, and gut microbiota of Chinese indigenous dogs. Fifteen Chinese indigenous dogs (6.34 ± 0.56 kg) were randomly assigned to three treatment groups and received either the control diet (CON), high-fat diet (HF), or high-fat diet containing 6% Mulberry leaf powder (MLP) for four weeks. The CON group received a basal diet, the HF group received a basal diet supplemented with 10% lard, and the MLP group received a basal diet supplemented with 10% lard and 6% MLP. The trial lasted for four weeks. The growth performance, lipid metabolism parameters, immune globulins, cytokines, and fecal microbiota were measured. Results showed that there was no significant difference in growth performance. The MLP group appeared to have decreased (*p* < 0.05) the serum level of low-density lipoprotein cholesterol (LDL-C) and apoliprotein-A1(APO-A1) in serum. The MLP group appeared to have higher (*p* < 0.05) serum immune globulin A (IgA) levels. UPGMA results showed that the MLP group was closer to the CON group than to the HF group. LEfSe analysis showed that dietary supplementation with MLP contributed to an alteration in the genus *Alloprevotella*, *Sarcina*, and species belonging to the *Bacteroides* and *Lactobacillus* genus. Overall, the dietary supplementation of 6% MLP can improve lipid metabolism conditions and immunity in high-fat-diet-fed dogs, and can alter the gut microbial composition of dogs.

## 1. Introduction

In a society where aging and oligonucleolization are becoming more prevalent [1], the importance of dogs as companion animals cannot be overstated. Dogs play an important role in people’s lives, and interaction with animals may positively impact their owners [2]. Previous research has shown that owning a dog can positively impact mental health by reducing feelings of loneliness and depression and increasing overall happiness and well-being [3]. Likewise, there have been a few studies highlighting the fact that having a pet has a certain positive effect on depression, anxiety, and behavioral disturbances [4,5,6]. Furthermore, for older adults who may be living alone, dogs can provide a sense of security and protection, and can also provide peace of mind [7]. Not only has it had an impact on psychology, but also on physical health. Dog owners exercise more and fall asleep more easily than non-dog owners [8]. Dogs may be beneficial in reducing cardiovascular risk in their owners by providing social support and motivation for physical activity [9,10].

Obesity is the most common nutritional disease which is characterized by the accumulation of adipose tissue to the point that health is adversely affected in dogs. When a dog is overweight or obese, it puts additional stress on its body and can lead to a variety of health problems, including joint problems, diabetes, respiratory problems, cardiovascular disease, and even shortened lifespan [11,12,13]. In people and animals, obesity is a multifactorial condition involving diet, a level of physical activity, behavioral factors, socioeconomic factors, environmental exposures, genetics, metabolism, and the microbiome [14]. Factors that cause obesity in dogs are related to the owners’ awareness of obesity-related health risks [15]. The primary cause of obesity in dogs is a combination of overfeeding and a lack of exercise. To prevent obesity in dogs, it is important to feed them a balanced diet, restrict feeding, provide plenty of exercise, and monitor their weight regularly [16].

Given the importance of dietary regulation in maintaining dogs’ health and preventing obesity, an increasing number of recent studies focused on investigating the effects of plant-derived bioactive components on dogs’ health. In recent years, mulberry leaves have been used in agricultural animals and have been shown to play important roles in regulating lipid metabolism, antioxidative capacity, immune function, and muscle development [17,18,19]. The existing body of research on mulberry leaf demonstrates that mulberry leaf exhibits a variety of biological activities, including antioxidant [20,21,22], anticancer [23], anti-obesity [24,25,26], anti-inflammation [26,27,28], and antibacterial [29] effects. Based on their beneficial biological activities, mulberry leaves are also used in Chinese traditional herbal medicine to treat diseases [30,31]. Furthermore, previous research has established that mulberry leaf can be a potential protein source or beneficial additive for domestic animal feeds [18,32,33]. In recent years, there has been a dramatic increase in the lipid-lowering effect of mulberry leaf powder. The effective components of mulberry leaf polyphenols inhibited adipogenesis through elevating leptin-stimulated lipolysis [34,35], ameliorating obesity through enhancing brown adipose tissue activity, and modulating gut microbiota [26]. The dietary fiber in mulberry leaves may be another critical factor regulating lipid metabolism in animals. High dietary fiber intake was shown to decrease the plasma total cholesterol (T Chol), triglyceride (TG), and free fatty acid (FFA) levels in rats [36]. In addition, the dietary fiber may interact with gut microbes and produce metabolites, like short-chain fatty acids (SCFAs), which can be absorbed into the circulation and affect metabolic regulation [37]. As gut microbiota is strongly associated with obesity [38], it is worth studying the effects of mulberry leaves on gut microbiota. Therefore, we hypothesized that the addition of mulberry leaf powder to the diet can affect lipid metabolism and intestinal flora composition in dogs. In this study, Chinese indigenous dogs were fed with a control diet, a high-fat diet (HF), or a high-fat diet containing 6% Mulberry leaf powder (MLP) for four weeks to investigate the impact of mulberry leaf powder (MLP) on growth performance, lipid metabolism parameters, immunity indicators, and gut microbiota.

## 2. Materials and Methods

The Animal Care and Use Committee of Guangdong Academy of Agricultural Sciences granted approval to the protocols and procedures governing the care and handling of animals in this study.

### 2.1. MLP Preparation

The mulberry leaves utilized in this study were acquired from the Southern China distribution of the National Germplasm Garden of Mulberry at the Guangdong Academy of Agricultural Sciences. After collection, the leaves were dehydrated via a drying apparatus, pulverized with a feed grinder, and sifted through 90 mesh sieves. The nutrient composition in mulberry leaf powders are as follows: crude protein: 14.0%, crude fat: 4.3%, crude fiber: 6.0%, crude ash: 8.3%, total calcium: 1.3%, and total phosphorous: 0.3%. The bioactive components in mulberry leaf powders are as follows: 1-deoxynojirimycin: 7.06 μg/g, gallic acid: 2.51 μg/g, protocatechuic acid: 9.44 μg/g, gallocatechin gallate: 0.14 μg/g, catechin: 0.04 μg/g, rutin: 657.88 μg/g, quercetin: 3.48 μg/g, chlorogenic acid: 3449.12μg/g, and naringenin: 0.35 μg/g.

### 2.2. Animals, Experimental Design, and Diets

The study involved fifteen 2-month-old Chinese indigenous dogs with similar initial body weight, which came from 4 litters of the same breeds. The animal study was conducted in the Huadu test base of Guangdong Academy of Agricultural Sciences. The 15 dogs were randomly assigned to three treatment groups, each comprising five dogs. The treatment groups were designated as follows: the control (CON) group, the high-fat (HF) group, and the mulberry leaf powder (MLP) group. The CON group received a basal diet, the HF group received a basal diet supplemented with 10% lard, and the MLP group received a basal diet supplemented with 10% lard and 6% MLP. The trial lasted for four weeks, during which all dogs had unlimited access to water and feed. Throughout the trial period, the initial body weight (BW), the body weight (at experimental days 7, 14, and 21), and the final BW of each dog were recorded, and the average daily gain (ADG) was calculated for each dog.

### 2.3. Sample Collection

At the end of the study, dogs were fasted overnight before blood was collected from their lateral saphenous vein. The collected blood was allowed to sit for 30 min before serum samples were separated by centrifugation at 3000× *g* and 4 °C for 10 min. The serum samples were then stored in −20 °C until analysis. In order to determine the changes in gut microbiota following different dietary treatment, the fecal samples were collected form each dog on the last day of the trial. The fecal samples were stored in −80 °C after they were snap-frozen until analysis.

### 2.4. Measurement of Lipid Metabolism Parameters

Triglycerides (TGs), total cholesterol (TCHO), high-density lipoprotein cholesterol (HDL-C), low-density lipoprotein cholesterol (LDL-C), Apolipoprotein-B (APO-B), and apoliprotein-A1 (APO-A1) were determined in serum collected from dogs in different groups at the end of the study using ELISA kits (Jiangsu Meimian Industrial Co., Ltd., Nanjing, China) in accordance with the manufacturer’s instructions.

### 2.5. Fecal Microbiota Analysis

The total DNA of each fecal sample was extracted using a DNA extraction kit (Omega Bio-Tek Inc., Norcross, GA, USA). After DNA extraction, specific primers with barcodes were synthesized based on the full-length primer sequences. The V3–V4 region of the bacterial 16S rRNA gene was amplified by PCR using primers 341F and 806R. When PCR amplification was performed, the products were purified, quantified, and normalized to form a sequencing library (SMRT Bell). The constructed library was first subjected to library quality control, and libraries that pass quality control were sequenced using PacBio Sequel. The circular consensus sequencing (CCS) sequences were obtained using the smrtlink tool provided by PacBio. Furthermore, lima v1.7.0 software was used to identify CCS sequences based on barcodes, generating raw CCS sequence data, and cutadapt 1.9.1 software was used to identify and remove adapter sequences and perform length filtering and generate clean CCS sequences without adapter sequences. Effective CCS sequences were generated using the UCHIME v4.2 software to identify and eliminate chimeric sequences. The effective CCS sequences were clustered/denoised to define OTUs/ASVs (referred to as “Features” hereafter), and their taxonomic classification was determined based on the sequence composition of features. At the 97% similarity level, the sequences were clustered into operational taxonomic units (OTUs) using USEARCH (version 10.0). Taxonomic analysis was performed on the sample at various classification levels. Alpha diversity analysis was conducted to investigate the species diversity within individual samples. Alpha diversity indices, including ACE, Shannon, Simpson, and Chao 1 indices, were calculated using QIIME2 software (version 2020.6). Beta diversity analysis was used to compare the differences in species diversity between different samples. Beta diversity was analyzed using Bray–Curtis distances and visualized using principal co-ordinates analysis (PCOA). Functional prediction analysis was performed to predict the gene functions or phenotypes of samples, as well as to calculate the abundance of functional genes or phenotypes.

### 2.6. Analysis of Immunity Indicators

Immune globulin A (IgA), immune globulin G (IgG), immune globulin M (IgM), interlukin-6 (IL-6), and tumor necrosis factor-α (TNF-α) in serum collected from dogs in different groups at the end of the study were determined using ELISA kits (Jiangsu Meimian Industrial Co., Ltd., Nanjing, China), in accordance with the manufacturer’s instructions.

### 2.7. Statistical Analysis

All data were firstly examined for normal distribution purposes and then analyzed via one-way ANOVA analysis in GraphPad Prism 8.0 (GraphPad Prism Software Inc., San Diego, CA, USA). Tukey’s test was performed to carry out post hoc testing. Differences between treatments were considered significant when *p* < 0.05. All data are expressed as means ± standard errors of the mean (SEM).

## 3. Results

### 3.1. Growth Performance

The growth performance results of the dogs throughout the entire experimental period are displayed in Figure 1 and Figure 2. It was found that, on experimental days 7, 14, 21, and 28, the BW of dogs in the HF and MLP groups was similar to that of the control group. Although the dogs in the MLP group seemed to have gained less body weight in the first half, second half, and overall experimental period, no statistical differences were detected between the groups.

### 3.2. Lipid Metabolism Parameters

In order to assess the effects of the dietary inclusion of MLP on lipid metabolism status in Chinese indigenous dogs, parameters reflecting the blood lipids, cholesterol, and their transportation were determined. As shown in Figure 3, the triglyceride (TG) and total cholesterol (TCHO) levels in serum were not different among the three treatment groups. Moreover, the dogs fed three different diets appeared to have similar serum concentrations of high-density lipoprotein cholesterol (HDL-C) and Apolipoprotein-B (APO-B). However, compared with dogs in the control group, dogs in the MLP group appeared to have decreased (*p* < 0.05) low-density lipoprotein cholesterol (LDL-C) levels in the serum. Additionally, the serum apoliprotein-A1(APO-A1) was significantly higher (*p* < 0.05) in HF dogs than in CON dogs, whereas MLP supplementation could significantly decrease (*p* < 0.05) the serum levels of APO-A1.

### 3.3. Immunity Indicators

As a traditional Chinese medicine, mulberry leaf has been considered to be capable of improving immunity. In order to assess the effects of the dietary inclusion of MLP on immune status in Chinese indigenous dogs, immune globulins and cytokines in blood were determined. As shown in Figure 4, in comparison to the dogs in the HF group, dogs in the MLP group appeared to have higher (*p* < 0.05) serum immune globulin A (IgA) levels. However, the serum levels of IgG and IgM were not different among the three groups. The serum concentrations of interlukin-6 (IL-6) and tumor necrosis factor-α (TNF-α) in the three groups were also similar.

### 3.4. Gut Microbiota Operational Taxonomic Unit (OTU) Analysis

As depicted in Figure 5, the control group had 196 OTUs, the HF group had 188 OTUs, and the MLP group had 175 OTUs. The number of shared OTUs between the CON and HF groups was 182, while there were 164 between the HF and MLP groups. Furthermore, the CON group and the MLP group had 169 common OTUs, and all three groups shared 161 common OTUs.

### 3.5. Species Annotation, Taxonomic Analysis, and Alpha Diversity Analysis

A naïve Bayes classifier combined with alignment was used to classify feature sequences based on the Silva reference database and obtain the taxonomic annotation for each feature. Then, the community composition of each sample at various taxonomic levels was assessed. At the phylum level, taxonomic information on the dominant species is shown in Figure 6. The dominant phylum species were *Firmicute*, *Bacteroidete*, *Proteobacteria*, *Fusobacteriota*, and *Fusobacteria*. Taxonomic information on the dominant species at the genus level comprised *Megamonas*, *Prevotella*, *Fusobacterium*, *Bacteroides*, and *Alloprevotella*. Alpha diversity is a major indicator illustrating the diversity of the gut microbiota. In this study, the ACE indices, Chao1 indices, Simpson indices, and Shannon indices were all calculated (Table 1), whereas none of these indices showed any difference among the groups.

### 3.6. Beta Diversity Analysis

To measure the degree of similarity between microbial communities, β-diversity was further evaluated using Bray–Curtis PCoA. In this study, PCoA was used to analyze the fecal samples of the dogs from different groups. The results showed that there was no significant separation among the three groups (Figure 7A). However, the UPGMA results showed that the MLP group was closer to the CON group than to the HF group (Figure 7B).

### 3.7. LEFSe Difference Analysis

To identify the differences in abundance between the groups, linear discriminant analysis (LDA) and LDA effect size (LEfSe) analysis were performed, and thee results are shown in Figure 8 with the histogram length indicating the contribution of various species. Notably, significantly higher numbers of 12 species belonging to the Firmicutes phylum, mainly Clostridium, were observed in the control group compared to the MLP group. Conversely, the MLP group exhibited a significant abundance of *Alloprevotella*, *Sarcina*, and *Lactobacillus*, which belong to the phylum *Bacteroidetes* and *Firmicutes*. Figure 8B presents the clade of different species, with nodes representing taxonomic levels ranging from the phylum to the genus or species. Red nodes signify the microbial taxa that played a crucial role in the control group, while green nodes represent those in the MLP group. In the control group, *Ruminiclostridium*, *Clostridium*, *Faecalitalea*, *Terrisporobacter*, and *Candidatus* were identified as crucial genera. In contrast, genera including *Alloprevotella*, *Bacteroides*, *Sarcina*, and *Lactobacillus* played a crucial role in the MLP group. However, no bacteria in the HF group were observed to play crucial roles compared to the other two groups.

## 4. Discussion

In today’s society, on the one hand, aging and oligonucleolization are becoming increasingly prevalent. On the other hand, the trend of young people choosing not to marry or have children has become mainstream. Therefore, the companionship of pets shows increased importance than in previous eras. More and more pet owners are regarding their pets as family members and thus are starting to pay more attention to the health of pets. Similar to humans, pets are also facing increasingly prominent “three highs” issues. A recent study showed that the overall prevalence rates of overweight and obesity in a dog group were 21.1% and 20.2%, which would contribute to a variety of disease processes and negatively affect the quality of life in dogs [39]. MLP has been recognized as a traditional Chinese medicine that could decrease fat deposition and improve health status. Previously, we have shown that dietary MLP could decrease backfat thickness and improve the lean percentage of pigs [17]. The case of whether dietary MLP could improve lipid metabolism and bring other beneficial outcomes in dogs is worthy of further research.

In this study, high-fat diets were given to dogs to create a hyperlipidemia model. The growth performance results indicated that, although dogs in the MLP group seemed to have slightly lower ADG in terms of numbers, no significant difference was observed among the three groups, which might be attributed to the fact that the feeding trial was not long enough to highlight the difference in growth rates. In comparison, the lipid metabolism indicators did present differences among the groups. The HF group tended to have higher LDL-C than the CON group. Compared with the high-fat-diet-fed dogs, the dogs in the MLP group appeared to have lower serum LDL-C concentrations. LDL-C is often called the bad cholesterol because it can build up on the arterial walls to initiate the formation of atherosclerotic plaques [40], raising the chances of health problems like a heart attack. Therefore, lowered LDL-C levels have beneficial effects on preventing cardiovascular diseases in MLP-fed dogs. Apo-A1, the major component of high-density lipoprotein, plays a crucial role in promoting reverse cholesterol transport by effectively incorporating phospholipids and cholesterol [41]. In this study, high-fat-diet-fed dogs appeared to have higher serum APO-A1 levels, while MLP consumption could decrease the level of APO-A1. A high-fat-diet-induced increase in APO-A1 has been proven in previous studies [42], and it may indicate that the body itself attempts to remove excess cholesterol. In the context of the dogs’ lipid profile, the difference in serum levels of LDL-C and APO-A1 in these groups reflects that MLP can improve lipid metabolism conditions in high-fat-diet-fed dogs.

The effects of mulberry leaf powder on regulating immunity in animal studies have also been well illustrated [43,44,45]. Herein, immune indicators were determined, and mulberry leaf consumption appeared to improve the serum levels of IgA. Immunoglobulin A (IgA), as a crucial component among the five major immunoglobulins, exerts a central role in maintaining mucosal homeostasis in the gastrointestinal, respiratory, and genitourinary systems [46]. It serves as the primary antibody in these contexts, governing an immune response. Therefore, MLP can improve dogs’ immunity through enhancing IgA secretion, whereas the underlying mechanisms need further research. Inflammatory cytokines directly reflect inflammatory status. In this study, the inflammatory cytokine levels in serum seemed to be the same among the three groups, indicating that the inflammatory status in different groups is similar.

In dogs, gastrointestinal health is also a very important issue that pet owners should pay attention to. Of note, the gut microbial environment is strongly associated with dogs’ body conditions [47]. The gut microbiota plays a pivotal role in extraintestinal disorders, including obesity [48]. Therefore, gut microbial composition was determined by 16S rRNA gene sequencing. In accordance with previous studies [49,50], Firmicutes and Bacteroidetes were identified as the predominant bacterial phyla in experimental dogs.

The UPGMA algorithm calculates the pairwise genetic distances between samples using a simple arithmetic mean and then groups the most similar samples together. According to the UPGMA results in this study, the MLP group was closer to the CON group than to the HF group, indicating a higher microbial community similarity between the MLP group and the CON group. LEfSe analysis showed that dietary supplementation with MLP contributed to an alteration in the genus *Alloprevotella*, *Sarcina*, and the species belonging to the Bacteroides and Lactobacillus genus. *Alloprevotella* is recognized as a beneficial bacterium and can produce SCFA and promote an anti-inflammatory environment. Moreover, *Alloprevotella* was reported to be negatively correlated with the abnormal lipid metabolism parameters, including serum LDL-C levels [51]. An alteration in the genus *Alloprevotella* might also partly be attributed to the effects of fibers on MLP. As shown in a previous study, mixed dietary fiber intervention for 4 days significantly promoted the growth of *Alloprevotella* in young healthy people [52]. Bacteroides has been recognized as a bacterial genus related to lipid metabolism. It was shown in a previous study that the abundance of Bacteroides was greatly reduced in patients with nonalcoholic fatty liver disease [53]. Some species of gut Bacteroides have been demonstrated to downregulate the expression of genes involved in lipogenesis and ameliorate lipid metabolic disorders [54,55]. Lactobacillus is a genus of bacteria that is considered to be beneficial and is commonly referred to as probiotics.

In a mice study, lactobacilli supplementation could significantly improve blood lipid levels and liver function and alleviate liver oxidative stress [56]. Moreover, lactobacilli supplementation significantly inhibited lipid accumulation in the liver and regulate lipid metabolism in epididymal fat pads. Lactobacillus species have also been shown to be associated with immunity regulation. In a pig study, Lactobacillus-treated pigs were found to have increased immunoglobin secretion and regulate immune indices [57]. Therefore, the altered gut microbiota in MLP-fed dogs may play important roles in the regulation of lipid metabolism and immunity.

## 5. Conclusions

In summary, dietary supplementation with 6% MLP decreased serum LDL-C and APO-A1 levels, increased the secretion of serum immune globulin A, and altered the gut microbial composition of dogs, whereas it had no effects on the growth rates. The improved lipid metabolism conditions and immune indices may be associated with the altered abundances of the bacteria genus. This study shows that dietary supplementation with 6% MLP can improve lipid metabolism conditions and immunity in high-fat-diet-fed dogs, and can also alter the gut microbial composition of dogs.

## Figures and Tables

**Figure 1 metabolites-13-00918-f001:**
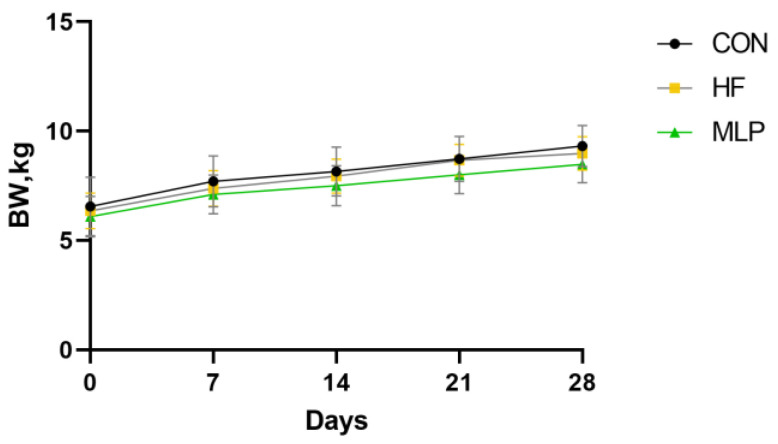
Effects of dietary supplementation with mulberry leaf powder (MLP) on the body weights of dogs. The body weights of all dogs were weighed and recorded every week. MLP, mulberry leaf powder; HF, high-fat dies; BW, body weight.

**Figure 2 metabolites-13-00918-f002:**
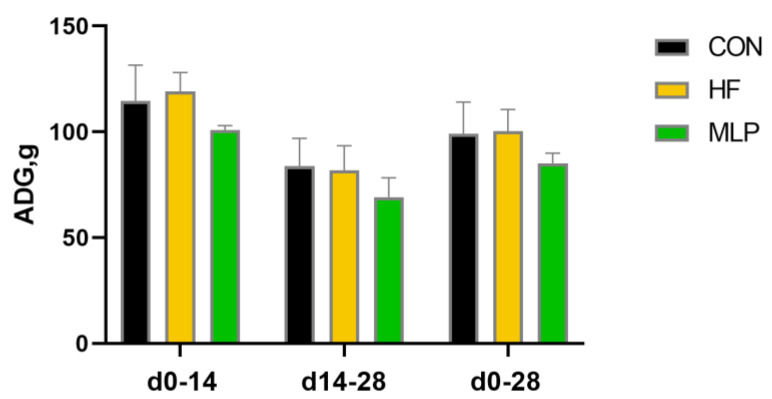
Effects of the dietary inclusion of MLP on average daily gain (ADG) of dogs during different experimental periods. MLP, mulberry leaf powder; HF, high-fat dies; ADG, average daily gain.

**Figure 3 metabolites-13-00918-f003:**
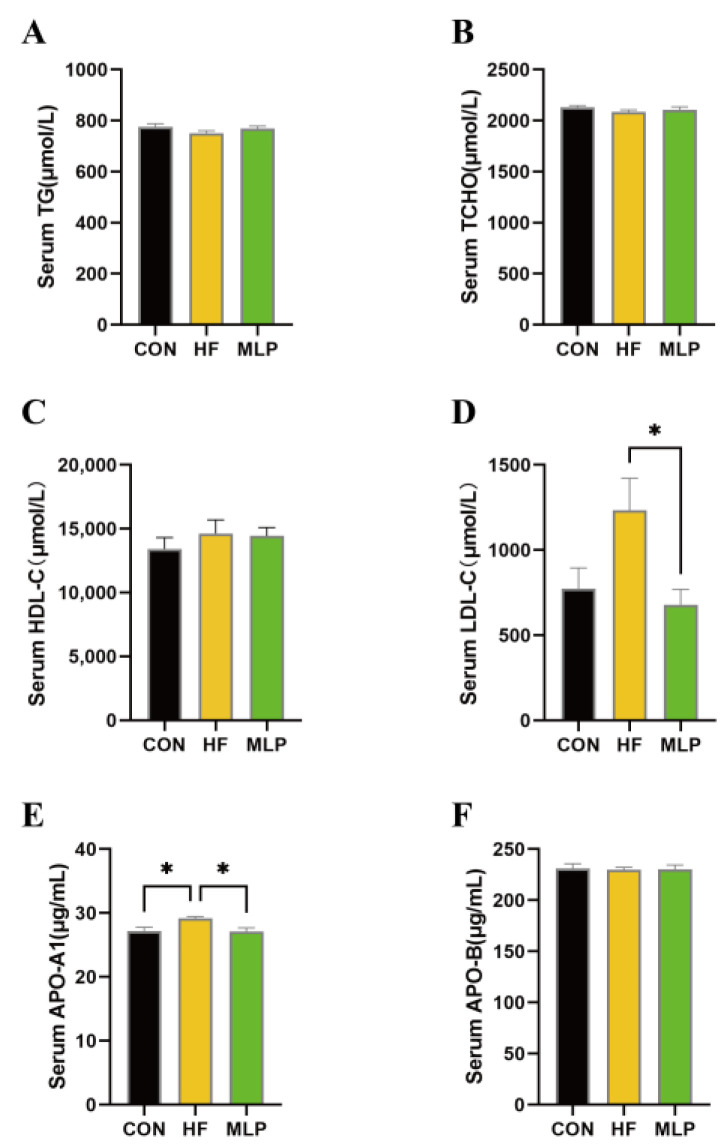
Effects of the dietary inclusion of MLP on lipid metabolism status in Chinese indigenous dogs. (**A**) Serum concentrations of triglycerides. (**B**) Serum concentrations of total cholesterol. (**C**) Serum concentrations of high-density lipoprotein cholesterol. (**D**) Serum concentrations of low-density lipoprotein cholesterol. (**E**) Serum concentrations of apoliprotein-A1. (**F**) Serum concentrations of Apolipoprotein-B. Data are expressed as means ± SEM. * indicates a significant difference (*p* < 0.05) between the two different groups.

**Figure 4 metabolites-13-00918-f004:**
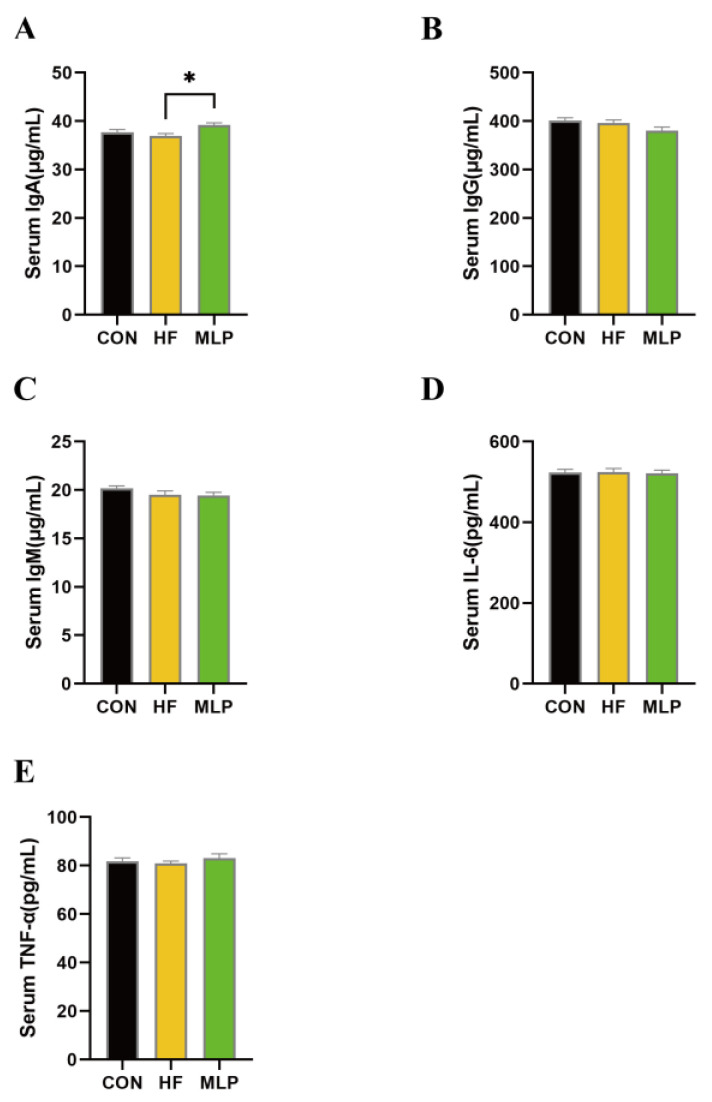
Effects of the dietary inclusion of MLP on immunity indicators in dogs. (**A**) Serum concentrations of immune globulin A. (**B**) Serum concentrations of immune globulin G. (**C**) Serum concentrations of immune globulin M. (**D**) Serum concentrations of interleukin-6. (**E**) Serum concentrations of tumor necrosis factor-α. Data are expressed as means ± SEM. * indicates a significant difference (*p* < 0.05) between two different groups.

**Figure 5 metabolites-13-00918-f005:**
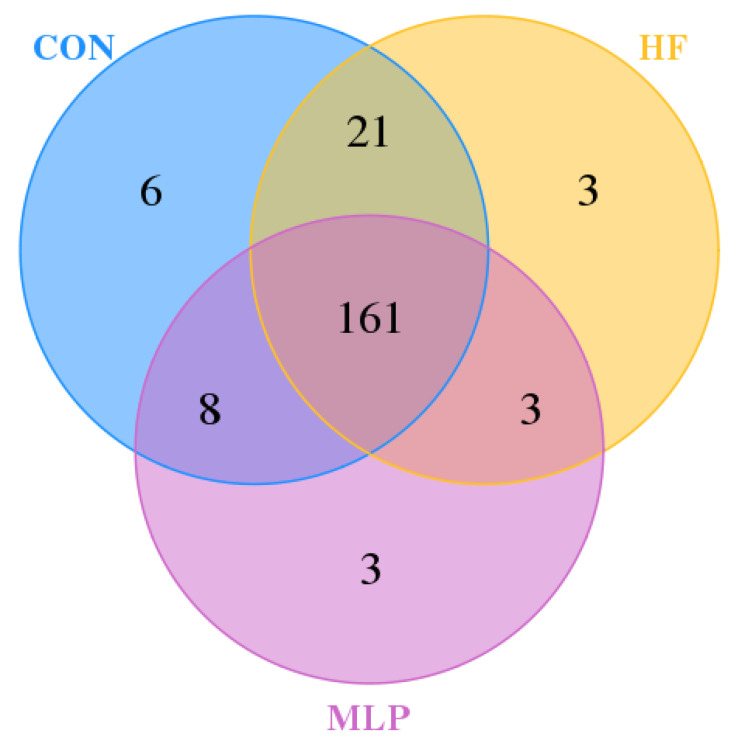
OTU Venn diagram.

**Figure 6 metabolites-13-00918-f006:**
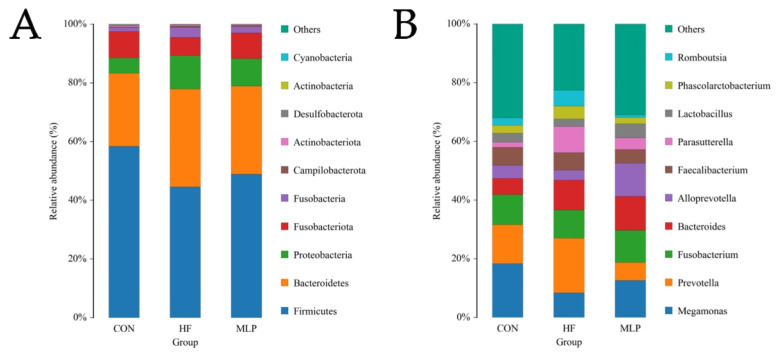
Phylum-level structural analysis. (**A**) Relative abundance of gut microbiota at the phylum level. (**B**) Relative abundance of the gut microbiota at the genus level.

**Figure 7 metabolites-13-00918-f007:**
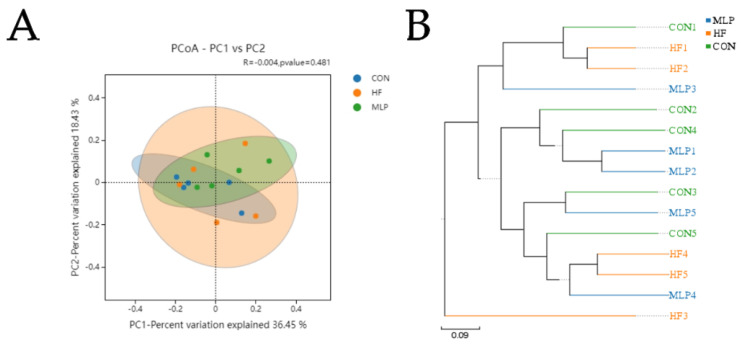
Dimension reduction analysis. Principal component analysis (PCA) for: (**A**) principal coordinate analysis (PCoA) of weighted UniFrac distances of 16S rRNA genes and (**B**) unweighted pair group method with arithmetic mean (UPGMA) analysis.

**Figure 8 metabolites-13-00918-f008:**
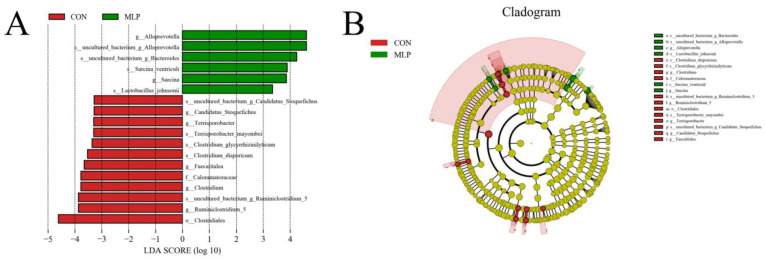
LEFSe difference analysis. (**A**) LDA scores for microbial taxa with significant roles in both groups. The LDA score represents the effect size. The histogram of the LDA scores presents species (biomarker) whose abundance showed significant differences between different groups. The LDA score represents the effect size. (**B**) LDA scores for microbial taxa with significant roles in both groups. In the cladogram, circles radiating from the inner side to the outer side represent taxonomic levels from the phylum to the genus (species). Each circle’s diameter is proportional to the taxon’s relative abundance.

**Table 1 metabolites-13-00918-t001:** Statistical analysis of microbial alpha diversity.

Group	ACE	Chao1	Simpson	Shannon
CON	149.92 ± 15.36	149.00 ± 15.72	0.85 ± 0.05	4.30 ± 0.45
HF	139.80 ± 7.94	136.10 ± 10.04	0.84 ± 0.03	3.87 ± 0.30
MLP	129.5 ± 7.38	129.2 ± 7.79	0.90 ± 0.01	4.33 ± 0.22

Data are presented as mean ± SEM. There was no significant difference among the treatment groups (*p* > 0.05).

## Data Availability

The data presented in this study are available on request from the corresponding author. The data are not publicly available because the authors are applying for patent.

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
