# Peer review of "Effects of Dietary Supplementation with Mulberry Leaf Powder on the Growth Performance, Lipid Metabolism Parameters, Immunity Indicators, and Gut Microbiota of Dogs"

_metabolites, 2023, doi:10.3390/metabo13080918_

Round 1

Reviewer 1 Report

Dear authors,

Your work is of very much interest as obesity is a significant concern in our days, both for humans and for company animals. Here you have a few comments to address:

1. at the Introduction and Discussion sections, please consider the other parameters that the diet supplementation with dietary fibers, such as hematological parameters, mineral absorption, and gut-microbial modulation could be directly influenced. Please consider amending these sections by discussing the following recent works on your topic: https://doi.org/10.31925/farmacia.2021.6.20, doi:10.15835/buasvmcn-vm:2020.0018, and 'Comparative ionogram assessment before and after probiotic treatment for healthy dogs and dogs with apparent dysbiosis - Lucrări Ştiinţifice Seria Medicină Veterinară, 63(1)/2020, USAMV Iaşi'.

2. were there any side effects of MLP administration in dog diets (eating appetite, constipation/diarrhea/no stool consistency modification, other significant changes)?

3. what is the chemical profile of the MLP powder in terms of dietary fibers, antioxidant compounds, and other compounds? Was it investigated?

4. what are the significant compounds associated with the cholesterol decrease and immunity stimulation triggered by MLP powder supplementation?

5. which fiber is associated with Alloprevotella stability?

Reviewer 2 Report

Dear authors,

The article entitled "Effects of Dietary Inclusion of Mulberry Leaf Powder on Growth Performance, Lipid Metabolism Parameters, Immunity Indicators and Gut Microbiota of Dogs" is a contribution to the question of how diet can help improve body condition and minimize obesity in dogs. To this end, the Chinese indigenous dogs were fed a control diet, a high-fat diet (HF) or a high-fat diet containing 6% mulberry leaf powder (MLP) for four weeks to investigate the effect of MLP supplementation on growth performance, lipid metabolism parameters, immunity indicators, and gut microbiota.

Line 20: 33.8 ±1.1. - the unit is missing. If it is the body weight, why does Table 1 say the body weight of about 5.5 kg? What about the age of the dogs and the BSC?

Introduction: there is too much about the human-animal relationship and too little about previous studies with MLP and facts. There are 33 lines on obesity and 10 on MLP. I believe the focus is on the possible use of MLP in the future. It should also be mentioned whether the concentration of different compounds in MLP varies depending on the time of sampling, drying temperature, growth area, etc.

Lines 98-100: What do you mean by this paragraph? The article under 19 says: "The results showed that mulberry leaves grown for 6 months had higher flavonoid content and that different drying methods could significantly affect the flavonoid content and antioxidant capacity of the leaves. So you need to be more specific about what material was used in your study. How do you explain the amount of 6% for the MLP?

Line 102: Please explain the term Chinese indigenous (native) dogs. Is the experiment about different breeds of dogs? What about age? What about the location of the experiment?

Chapter 3.1: The results show only average values of body weight. Did body weight and food intake differ among dogs in the same group? What was the difference in baseline body weight between the CON and MLP groups? What about the consistency and quantity of feces? In the HF group, the average body weight stagnated between 21 and 28 days. Do you have any explanation for this?

Figure 2: Do you have an explanation for why the ADG is lowest in the MLP group?

3.2: Use capital letters

Line 207: Figure 5 instead of Figure 10.

Line 225: Why do you use the term prebiotic groups?

Line 276: reference is missing

Line 321: Explain SCFA

Please be more specific in the text - use the abbreviation MLP after the first explanation.

Author Response

Please see the attachmentPlease see the attachment

Round 2

Reviewer 1 Report

The authors improved the quality of their manuscript.

Reviewer 2 Report

Dear authors,
thank you for your well-prepared answers and for improving the text.
I
have no further questions.